# Higher Disease and Pain Severity and Fatigue and Lower Balance Skills Are Associated with Higher Prevalence of Falling among Individuals with the Inflammatory Disease of Neuromyelitis Optica Spectrum Disorder (NMOSD)

**DOI:** 10.3390/jcm9113604

**Published:** 2020-11-09

**Authors:** Mahdi Barzegar, Dena Sadeghi Bahmani, Omid Mirmosayyeb, Reyhaneh Azarbayejani, Alireza Afshari-Safavi, Saeed Vaheb, Nasim Nehzat, Afshin Dana, Vahid Shaygannejad, Robert W. Motl, Serge Brand

**Affiliations:** 1Isfahan Neurosciences Research Center, Alzahra Research Institute, Isfahan University of Medical Sciences, Isfahan 81746-73461, Iran; barzegar_mahdi73@yahoo.com (M.B.); omid.mirmosayyeb@gmail.com (O.M.); reyhazar@gmail.com (R.A.); saeedvaheb.sv@gmail.com (S.V.); n.nehzat96@gmail.com (N.N.); sin.afshin@yahoo.com (A.D.); 2Department of Neurology, Isfahan University of Medical Sciences, Isfahan 81746-73461, Iran; 3Departments of Physical Therapy, University of Alabama at Birmingham, Birmingham, AL 35209, USA; dena.sadeghibahmani@upk.ch (D.S.B.); robmotl@uab.edu (R.W.M.); 4Center of Affective, Stress and Sleep Disorders (ZASS), University of Basel, Psychiatric Clinics (UPK), 4002 Basel, Switzerland; 5Sleep Disorders Research Center, Kermanshah University of Medical Sciences (KUMS), Kermanshah 67198-51351, Iran; 6Student Research Committee, School of Medicine, Isfahan University of Medical Sciences, Isfahan 81746-73441, Iran; 7Department of Biostatistics and Epidemiology, Faculty of Health, North Khorasan University of Medical Sciences, Bojnurd 74877-94149, Iran; alireza.afsharisafavi@gmail.com; 8Substance Abuse Prevention Research Center, Health Institute, Kermanshah University of Medical Sciences (KUMS), Kermanshah 67198-51351, Iran; 9School of Medicine, Tehran University of Medical Sciences (TUMS), Tehran 14117-13135, Iran; 10Department of Sport, Exercise, and Health, Division of Sport Science and Psychosocial Health, University of Basel, 4051 Basel, Switzerland

**Keywords:** inflammatory disease, neuromyelitis optica spectrum disorder, falling, predictors, pain, fatigue, balance

## Abstract

Background: Neuromyelitis optica spectrum disorder (NMOSD) is a chronic inflammatory and autoimmune disorder that is associated with impaired vision, sensory loss, pain, fatigue, and spasms in the upper and lower limbs. Typically, persons with this disorder are also at higher risks of falls. Given this, the aims of the study were to compare the prevalence rates of falling for NMOSD cases and healthy controls (HCs), and to predict falling in the former group based on sociodemographic, psychological, and illness-related factors. Method: A total of 95 adults with NMOSD (Mean age = 34.89 years; 70.5% females) and 100 matched HCs took part in the study. All participants completed a series of questionnaires covering sociodemographic information and falling rates. The NMOSD individuals also reported on disease duration, pain, fatigue, and fear of falling, while their balance performance was objectively assessed. Results: Compared to healthy controls, the NMOSD cases had a 2.5-fold higher risk of falling. In this latter group, higher scores for pain, fatigue, fear of falling, and higher EDSS scores were distinguished between fallers and non-fallers, and objective balance skills had no predictive value. Conclusions: Compared to healthy controls, NMOSD sufferers had a 2.5-fold higher risk of experiencing falls. In this group, disease impairments (EDSS, fatigue, pain) predicted falling. Specific interventions such as regular resistance training might reduce the risk of falling.

## 1. Introduction

Neuromyelitis optica spectrum disorder (NMOSD/NMO) is a chronic inflammatory disorder characterized by impairments of the optic nerve, the spinal cord, and the brain [1,2,3]. Often, persons with NMOSD suffer from (a) visual impairment/decreased visual acuity, and (b) spinal cord dysfunction (muscle weakness, reduced sensation, loss of bladder and bowel control), along with an acute and severe spastic weakness of the legs (paraparesis), and in some cases, all four limbs (quadriparesis) [3,4,5]. Given this, impairments in gait and falling would be unsurprising.

However, a thorough literature search using the most comprehensive search engines PubMed^®^, Google Scholar^®^, and Scopus^®^ (end of August 2020) showed that while impairments in gait were described in three case studies [6,7,8], no study has so far focused on falling associated with this disorder. This is surprising for three reasons. First, as mentioned, severe spastic weakness of the legs is a core symptom of NMOSD and one would therefore expect impaired gait and higher rates of falling to be features of this disorder. Second, higher frequency and prevalence rates of falling are well documented among persons with multiple sclerosis (MS) [9,10,11,12,13,14]. Third, NMOSD has been identified as a neurological disorder distinct from MS only within the last 15 to 20 years [15,16,17,18]. Specifically, the detection of pathogenic antibodies to aquaporin-4, a water channel present in astrocytic foot processes mainly expressed in the optic nerves, brainstem, and spinal cord, has helped to distinguish NMOSD from MS [2,15,16,17,18]. Fourth, reviews of falling among persons with MS have shown poor cognitive processing (verbal memory), particularly associated with impaired gait and increased risks of multiple falls [19]. Surprisingly, other illness-related factors such as the severity of disease (as measured with the Expanded Disability Status Scale [EDSS]), physical activity, fatigue, and illness duration did not appear to be linked to falling, though it may also be the case that such possibilities have not so far been considered.

To summarize, given the neurological and musculoskeletal impairments observed among NMOSD sufferers, one would also expect higher rates of falling than in healthy controls.

The following two hypotheses and one research question were formulated. First, as with multiple sclerosis (MS) sufferers [9,10,11,12,13,14], we expected that those with NMOSD would report more frequent falls than healthy subjects. Second, following D’Orio et al. [19], we anticipated that more marked cognitive impairments would be associated with higher fall rates. We treated as exploratory the question of the extent to which typical NMOSD-related factors such as higher scores on the EDSS and for fatigue, and pain, together with lower balance skills, might predict the frequency of falls.

We believe that the result from this study may pave the way to further investigation of the significance of falling in NMOSD. Depending on the pattern of results, there may also be implications for specific therapeutic interventions.

## 2. Methods

### 2.1. Procedures

Between September 2017 and October 2019, adults suffering from NMOSD and attending the multiple sclerosis clinic of the Kashani Hospital in Isfahan (Isfahan University of Medical Sciences, Isfahan, Iran) were approached to participate in the study. In parallel, sex- and aged-matched controls were recruited. All participants were informed about the study aims and the secure and anonymous data handling. Thereafter, they all signed a written informed consent. All participants completed a series of questionnaires covering sociodemographic and falling-related information. The NMOSD sufferers additionally completed questionnaires covering illness-related information such as disease duration, fatigue, and pain. Last, their balance skills were objectively assessed (see below). The local ethical committee approved the study (IR.MUI.REC.1396.1.184), which was conducted in accordance with the ethical principles laid down in the seventh revision [20] of the Declaration of Helsinki.

### 2.2. Participants

#### 2.2.1. Persons with Neuromyelitis Optica Spectrum Disorder (NMOSD)

Inclusion criteria were as follows: (1) Outpatient and suffering from NMOSD, as ascertained by a thorough neurological assessment in accordance with the international consensus diagnostic criteria [21] and performed by an experienced neurologist; (2) Age between 18 and 50 years. Given that increasing age may be associated with deteriorating gait and more frequent falls, the age limit was set at 50 years; (3) EDSS score between 0 and 6 (“6.0: Intermittent or unilateral constant assistance (cane, crutch or brace) required to walk 100 m with or without resting”); (4) Willing and able to comply with the study conditions, specifically, completing questionnaires in Farsi; and (5) Signed written informed consent. Exclusion criteria were: (1) Other neurological disorders such as Parkinson’s disease, multiple sclerosis, clinically isolated syndrome, or Alzheimer’s disease; (2) Psychiatric disorders such as major depressive disorders, substance use disorder, anxiety disorders, or personality disorder; (3) Musculoskeletal injuries due to accidents and similar injuries within the last 12 months; (4) Relapse within the last six months as a recent relapse could have had an impact on gait quality, falling, and fear of falling, which was the reason for excluding these cases; (5) Uncertain NMOSD diagnosis; and (6) Myelin oligodendrocyte glycoprotein antibody disease (MOGAD).

#### 2.2.2. Healthy Controls

Healthy controls were recruited via flyers and advertisements among hospital staff members. Inclusion criteria were: (1) Age between 18 and 50 years; (2) Able to comply with the study conditions; and (3) Signed written informed consent. Exclusion criteria were: (1) Known neurological and/or psychiatric disorders as defined above; and (2) Musculoskeletal injuries due to accidents and similar injuries within the last 12 months.

### 2.3. Tools

#### 2.3.1. Sociodemographic and Anthropometric Information

All participants reported on their age, sex, educational level (diploma vs. higher education) height, and weight.

#### 2.3.2. Disability Status and Illness Duration

For NMOSD patients, an experienced neurologist rated the degree of impairment using the EDSS (Expanded Disability Status Scale; [22]). Illness duration was taken from medical records.

#### 2.3.3. Falling Incidence and Prevalence

To assess incidence and prevalence of falling, participants answered the following question: “In the last six months, did you fall down? Falling means: Unexpectedly, and despite your efforts, you lost equilibrium and fell to the ground or floor.” The answer was “yes” or “no”. If yes: “How many times did you fall in the last six months?”. Answers were: “once”, “twice”, or “more than twice”. We followed Gunn et al. [12] and dichotomized NMOSD patients into fallers (falling two or more times within the last six months) and non-fallers (not falling at all or falling once within the last six months).

#### 2.3.4. Fear of Falling: Fall Efficacy Scale

To assess fear of falling, the Iranian version [23] of the Falls Efficacy Scale International (FES-I) [24] was employed. The self-rating questionnaire consists of 16 items. Typical items are: “going grocery shopping”, “taking a bath or a shower”, “meeting friends or relatives”, or “attending an event such as family parties, religious rituals, etc.”. Answers were given on 4-point rating scale ranging from 1 (not at all concerned) to 4 (very concerned), with higher sum scores reflecting a greater fear of falling (Cronbach’s alpha = 0.9).

### 2.4. Pain

To assess pain, NMOSD patients completed the Iranian version [25] of the Brief Pain Inventory (BPI) [26]. The self-rating questionnaire consists of several 10-point Likert scales. Typical items are: “On a scale from 1 to 10, indicate the highest pain severity of the last 24 h”, “On a scale from 1 to 10, indicate the lowest pain severity of the last 24 h”, “On a scale from 1 to 10, indicate the currently highest pain severity”, “On a scale from 1 to 10, indicate, if and how much pain has impacted on your mood/sleep/social contacts”, where 1 equals “not at all”, and 10 equals “almost unbearable/extremely high”; some items were reversed scored where a higher sum reflects greater pain severity.

### 2.5. Fatigue

To assess their level of self-perceived fatigue, participants completed the Fatigue Severity Scale (FSS; [27]). The Farsi version was validated [28] and administered to MS sufferers in a previous study [29]. The FSS consists of nine items, and answers are given on seven-point rating scales ranging from 1 (not at all) to 7 (definitively/almost always), with higher scores reflecting higher levels of fatigue. Previous research has shown that the FSS can be considered as a valid and reliable tool to assess respondents’ fatigue [30]. In the present sample, the Cronbach’s α was 0.96.

### 2.6. Balance

To provide an objective assessment of balance, NMOSD patients underwent the procedure based on the Berg Balance Scale ([31]; Farsi version: [32]). The BBS consists of 14 activities. The participant was asked to perform these activities, and the assessor rated the performance on a 5-point Likert scale ranging from 4 (fully accomplished) to 0 (not at all accomplished). A typical task is: The participant is sitting on a chair; then they are asked to stand up. Four points are given when the participant stands up without using their hands to push or stabilize. Three points are given when the participant uses their hands. Two points are given when they need two or more trials to stand up. One point is given when they need minimal help from another person, and zero points are given when the participant needs moderate to maximum help. In another task, the participant stands still and upright for two consecutive minutes. Four points are given when they fully accomplish the task. Three points are given when they move a foot/feet once or twice to another position; two points are given when they need help once. One point is given when the participant moves repeatedly and needs help. Zero points are given when they are unable to stand still and upright without moderate to maximum help. The sum score ranged from 0 to 56 points, with higher scores reflecting better balance.

### 2.7. Cognitive Impairment; Mini-Mental Sate Examination

Cognitive impairment was assessed with the Mini-Mental State Examination (MMSE [33], a widely applied expert rating instrument to screen for cognitive impairment. As described elsewhere in more detail [34], the screening instrument consists of 30 items and enables an estimate of the severity of cognitive impairment concurrently and longitudinally. The MMSE is considered an efficient and effective way to assess cognitive change during treatment. Specifically, it assesses various cognitive functions such as short- and long-term memory, orientation, and arithmetic. Lower scores reflect greater cognitive impairment.

## 3. Aquaporin-4 Immunoglobulin G (AQP4-IgG)

All patients were tested for AQP4-IgG with the cell-based indirect immunofluorescence assay [35].

## 4. Statistical Analyses

To compare sociodemographic and anthropometric dimensions of NMOSD patients and healthy controls, *t*-tests and X^2^-tests were performed.

An odds ratio was performed to determine whether falling (yes vs. no) was greater among NMOSD patients than healthy controls.

For the NMOSD patients only, *t*-tests and X^2^-tests were computed to compare sociodemographic, anthropometric, and disease-related dimensions between fallers and non-fallers. To predict who in this group reported falling, a logistic regression analysis was performed. Due to the multi-collinearity between variables, pain, Berg balance scale score, fatigue severity, mini-mental state examination, and fear of falling were considered as potential predictors while sex, age, education, current EDSS, disease duration, and BMI were treated as potential confounders. First, a non-adjusted analysis was conducted to determine the influence of each predictor variable on falling. Then, an adjusted model was developed by considering potential confounders. A backward stepwise selection was used to obtain a final model with the most important predictors that were significantly associated with falling. The order of variable selection was determined by an evaluation of the Wald statistic. Finally, the performance of adjusted logistic regression models was assessed using the area under the curve (AUC) of the receiver-operating characteristics and measures of diagnostic accuracy including sensitivity, specificity, positive predictive value, and negative predictive value. The optimal cut-off point of the model was determined using the Youden index [36], calculated as (sensitivity + specificity − 1). The results of the logistic regression analyses were recorded as odds ratio (OR), 95% confidence interval (CI), Wald statistic, degree of freedom, and *p*-value.

For t-tests, effect sizes were reported in terms of Cohen’s d.

The nominal level of significance was set at alpha < 0.05. All computations were performed with SPSS^®^ 25.0 (IBM Corporation, Armonk, NY, USA) for Apple Mac^®^.

## 5. Results

### 5.1. Sociodemographic and Illness-Related Information between Persons with NMOSD and Healthy Controls

Table 1 reports the descriptive and inferential statistical indices for the sociodemographic of NMOSD patients and healthy controls. Note that all statistical indices are reported in Table 1; accordingly, statistics are not repeated in the text.

The NMOSD patients and healthy controls did not differ with regard to age, sex, or educational level.

For NMOSD patients, positive AQP 4-Ab was observed in 34 out of 95 participants. About 12% used rituximab as a disease-modifying therapy.

### 5.2. Odds of Falling in NMOSD Patients and Healthy Controls

Table 2 reports the odds ratio calculations.

The odds of being a faller was 2.49-fold higher for NMOSD patients than for healthy controls.

### 5.3. Sociodemographic and Illness-Related Dimensions between Fallers and Non-Fallers among NMOSD Patients

Table 3 reports the descriptive and inferential statistical indices for sociodemographic and illness-related dimensions of fallers and non-fallers.

Compared to non-fallers, fallers had higher EDSS scores, a longer disease duration, higher pain scores, lower balance capacities, higher fatigue scores, and a greater fear of falling. Fallers and non-fallers did not differ with respect to sex ratio, age, education, AQP4-Ab, BMI, or cognitive impairment (MMSE).

### 5.4. Predicting Falling in NMOSD Patients

Falling (yes vs. no) was the outcome variable, and sex, age, education, EDSS, disease duration, BMI, pain, balance, fatigue, fear of falling, cognitive impairment were predictors. Table 4 gives a statistical summary. Two calculations were performed. In the non-adjusted analysis, sex, age, education, current EDSS, disease duration, and BMI were entered as independent predictors. In the adjusted analysis, these dimensions were controlled for.

Higher pain scores, higher fatigue, and greater fear of falling predicted falling status, while balance skills, and cognitive impairments were excluded from the equation as they did not reach statistical significance.

The performance of the adjusted models was further assessed using analysis of the ROC curve for pain, FSS, and FES (Table 5). The area under the curve was 0.974 for pain, 0.962 for FSS, and 0.994 for FES, respectively. In addition, using the Youden index, sensitivity and specificity were 0.923 and 0.774 for pain, 0.884 and 0.934 for FSS, and 0.964 and 1 for FES, respectively.

## 6. Discussion

The key findings of the present study were that, compared to healthy controls, patients with neuromyelitis optica spectrum disorder (NMOSD) had a 2.5-fold higher risk of falls within the last six months. Furthermore, higher scores for fear of falling, pain, and fatigue, along with a higher EDSS and poorer balance skills were the key features distinguishing fallers from non-fallers. Finally, against expectations, cognitive impairments had no predictive value. The present data add to the current literature in an important way in that, to our knowledge, this is the first study to assess the issue of falling among NMOSD sufferers in a thorough and scientific fashion. The present data are of practical importance in that falling in this group needs special attention because it may be associated with other physical injuries and with behavior changes to avoid daily physical activity. However, there is sufficient evidence that avoidance of physical activity may lead to additional psychological and physical complaints. Given these risks, there is a case for arguing that NMOSD patients should, in particular, be trained to improve their skills in balance, physical activity, and daily walking activities to reduce the likelihood of avoidance behavior and a progressive reduction in their physical independence.

Two hypotheses and one research question were formulated and these are now considered in turn.

Our first hypothesis was that, compared to healthy controls, NMOSD patients would report more frequent falls within the last six months, and this was confirmed; the risk in the patient group of being categorized as a faller was 2.5-fold higher. NMOSD sufferers are therefore in this respect similar to persons with MS [9,10,11,12,13,14]. The present study adds to the current literature in being the first to investigate this research question, one which may have clinical and practical implications.

Our second hypothesis was that NMSOD patients would be similar to MS sufferers in that cognitive impairments would predict the occurrence of falling; this hypothesis was not supported. Accordingly, the present findings are at odds with those reported for persons with MS [19]. We believe the following may explain this difference. First, in the present study, we included a broad range of other potential predictors of falling (see below). However, as is apparent in Table 3, the effect size for difference in MMSE scores between fallers and non-fallers was small and therefore negligible, even with no additional predictors entered in the equation. Second, this instrument (the MMSE; [33]) for assessing cognitive impairment by definition provides a rough and coarse-grained screening for cognitive functioning. Given this, it is possible that more precise assessments of working memory, long-term memory, perception, language, decision making, and problem solving [37] could have yielded a more differentiated pattern of results.

Next, we treated as an exploratory research question the extent to which typical NMOSD-related factors such as higher scores on EDSS, fatigue, and pain, and lower balance skills predict the occurrence of falling among NMOSD patients. As shown in Table 2, Table 3 and Table 4, the pattern of results is complex, but it emerges that falling is not associated with the current state of impairment or with objective balance skills. Rather, higher pain, higher fatigue, and greater fear of falling predicted falling status (see Table 4 and Table 5). To put it another way, falling appears to be a matter of cognitive-emotional processes and not a question of objective impairment of motor-related balance skills. This conclusion is reinforced by our finding that illness duration had no additional predictive value (Table 4), despite the fact that compared to non-fallers, fallers had almost twice the length of illness duration (Table 3).

Given these results, one might ask what if any treatment intervention might reduce the prevalence of falling.

For the following four reasons, we would encourage NMOSD patients to increase the intensity and frequency of their physical activity. First, there is some evidence that regular physical activity has a positive impact on pain perception. While an overview of 21 Cochrane reviews, covering 381 studies and 37,143 participants, concluded that the evidence linking physical activity and exercise to chronic pain is weak [38], such overviews of reviews also carry the risk that more fine-grained results are overlooked. There have been studies showing a clear reduction in pain severity and improved physical function. However, the authors of the overview also stressed that these studies mostly reported small-to-moderate effects and were not consistent across the reviews. One systematic review and meta-analysis showed that regular leisure time physical activity could reduce the risk of chronic low back pain by 11–16% [39]. Given these results, future intervention studies might assess whether or to what extent regular physical activity positively influences pain perception among NMOSD patients.

Second, for MS sufferers [40,41], there is good evidence that regular physical activity has a positive impact on fatigue. One might anticipate a similar benefit of physical activity interventions for those with NMOSD.

Third, with regard to the influence of regular physical activity on anxiety, the results are mixed [42,43]. With regard to fear of falling, evidence has been gathered from older people living in their communities [44]. For individuals with Parkinson’s disease, dancing, water exercise, and robotic gait training all appeared to be effective in improving balance and gait. For want of equivalent evidence for NMOSD sufferers, one might speculate that regular physical activity interventions could also have beneficial effects for balance, gait, and, ultimately, fear of falling.

Fourth, regular physical activity has been shown to improve self-esteem [45,46], at least in non-clinical samples of children and adolescents. One might expect that there would be the same effect of such activity for NMOSD sufferers. Furthermore, it is possible there would be a knock-on benefit of improved self-esteem through reduced fear of falling.

The novelty of the results should be balanced against the following limitations. First, as for all studies of falls, falling or near-falling can be interpreted in different ways. Consequently, incidence and prevalence rates may be under- or over-overestimated. Second and relatedly, we dichotomized falling status into “yes” or “no” within a time interval of six months, while a further differentiation of the incidence and prevalence rates and a longer time frame might have yielded a more complex and fine-grained pattern of results. Third, only NMOSD patients willing and able to comply with the study conditions participated; it follows that the sample could be biased. Fourth, other latent and unassessed physiological and psychological variables might have biased two or more dimensions in the same or opposite directions. In particular, vision accuracy and vision correction were not systematically assessed. It follows that future studies should take into consideration the quality of vision. Fifth, greater overall physical fitness and muscle strength are related to less frequent falls, at least among older people [47,48,49]. In the same vein, regular exercising to improve muscle strength has been found to be associated with better physical stability and lower incidence and prevalence rates of falling, again at least among older people [50,51,52,53]. Sadly, in the present study, we missed the opportunity to assess physical activity levels either subjectively or objectively. Given this, it is possible that indices of physical activity and exercise might have biased the present pattern of results. Furthermore, to our knowledge, no such interventional studies have so far been performed with MNOSD patients. A promising approach to the reduction of risks of falls in this group would be to systematically improve muscle strength. Sixth, the cross-sectional nature of the study precludes detection of causal relationships and, strictly speaking, regression analyses are therefore speculative claims about causes. Given this, an opposite or bi-directional relationship appears to be more plausible (though, not testable with the present data), one in which falling might trigger insecurity, distress, and avoidance behavior, which in turn, decrease confidence in the capacity to move, walk, and stay physically active. Such avoidance behavior may increase the risk of falling and of worsening overall physical status as reflected in higher EDSS scores. In brief, it is entirely possible that falling and physical and psychological functioning are related in a bi-directional fashion. Last, as with studies of MS [12,54,55], there should be an investigation of interventions to improve the muscle strength of people with NMOSD.

## 7. Conclusions

Compared to healthy sex- and aged-matched controls, persons with NMOSD reported more frequent falls. In this group, falling was associated with higher scores for pain, fatigue, and fear of falling, but not with poorer cognitive performance.

## Figures and Tables

**Table 1 jcm-09-03604-t001:** Demographic and clinical characteristics.

Variables	NMOSD	Healthy Control	Statistics
N	95	100	
	M (SD)	M (SD)	
Age, years	34.89 (9.41)	35.99 (9.02)	t(193) = 0.81, d = 0.11 (T)
	n/n (%/%)	n/n (%/%)	
Sex; female/male	67/28 (70.5/29.5)	79/21 (79/21)	X^2^(N = 195, df = 1) = 1.86
Education (diploma/higher diploma)	44/51 (46.3/53.6)	65/35 (35/65)	X^2^(N = 195, df = 1) = 0.80
			-
AQP 4-Ab; positive, n (%)	34 (35.7)	-	-
Disease modifying therapy (rituximab/other)	11/84 (11.6/88.4)	-	-

Notes: NMOSD = neuromyelitis optica spectrum disorder; T = trivial effect size.

**Table 2 jcm-09-03604-t002:** Odds ratio calculations for being a faller or non-faller, NMOSD patients versus healthy controls.

	Faller	Non-Faller	Non-Adjusted		Adjusted by Sex and Age	
			OR (95% CI)	*p*-Value	OR (95% CI)	*p*-Value
Patients; n (%)	85 (85)	15 (15)	3.02 (1.51–6.03)	0.002	2.50 (1.22–5.12)	0.013
Control; n (%)	33 (34.7)	62 (65.3)				

**Table 3 jcm-09-03604-t003:** Descriptive and statistical comparisons of sociodemographic and illness-related information between individuals with neuromyelitis optica spectrum disorder (NMOSD) labelled as fallers and non-fallers.

	Groups		Statistics
Dimensions	Fallers	Non-Fallers	X^2^-Tests
N	33	62	
	n (%)	n (%)	
Sex; female	23 (69.7%)	44 (71%)	X^2^(N = 95, df = 1) = 0.02
Education; Higher-diploma	17 (68%)	34 (54.8%)	X^2^(N = 95, df = 1) = 1.27
AQP 4-Ab; positive	12 (36.4%)	22 (35.5%)	X^2^(N = 95, df = 1) = 0.45
	M (SD)	M (SD)	*t*-tests
Age; mean (SD)	36.85 (9.18)	34.03 (9.45)	t(95) = 1.30, d = 0.30 (S)
Disease duration; years	6.57 (5.14)	3.95 (2.77)	t(95) = 3.10 **, d = 0.63 (M)
BMI	24.09 (4.18)	24.48 (4.03)	t(95) = 0.40, d = 0.09 (T)
Pain	4.69 (1.52)	2.49 (1.12)	t(95) = 8.02 ***, d = 1.65 (L)
Balance score (BBS)	40.30 (11.09)	52.64 (3.02)	t(95) = 8.02 ***, d = 1.52 (L)
Fatigue (FFS)	52.73 (11.64)	35.86 (6.99)	t(95) = 8.99 ***, d = 1.76 (L)
Fear of falling (FES-I)	48.25 (7.92)	21.50 (4.53)	t(95) = 18.77 ***, d = 4.15 (L)
Mini Mental State Examination	29.54 (1.12)	28.82 (2.12)	t(95) = 1.82, d = 0.42 (S)
	Median (range)	Median (range)	U-test
Current EDSS	3 (0–6)	1 (0–2)	Z = 6.26 ***

Notes: EDSS = expanded disability status scale; BBS = Berg balance scale, FSS = fatigue severity scale; MMSE = mini mental state examination; FES-I = Falls Efficacy Scale International. ** = *p* < 0.01; *** = *p* < 0.001; T = trivial effect size; S = small effect size; M = medium effect size; L = large effect size.

**Table 4 jcm-09-03604-t004:** Non-adjusted and adjusted analysis of risk factors affecting the development of falling in NMOSD patients.

	Non-Adjusted Analysis				Adjusted Analysis			
	OR (95% CI)	Wald	df	*p*	OR (95% CI)	Wald	df	*p*
Pain	3.26 (2.06–5.16)	25.41	1	<0.001	2.11 (1.21–3.66)	6.98	1	0.008
BBS	0.81 (0.74–0.89)	18.58	1	<0.001	0.91 (0.75–1.09)	1.16	1	0.282
FSS	1.21 (1.12–1.30)	25.46	1	<0.001	1.28 (1.05–1.56)	6.18	1	0.013
MMSE	1.30 (0.96–1.76)	2.965	1	0.085	1.55 (0.69–3.46)	1.12	1	0.290
FES	1.76 (1.12–2.76)	6.07	1	0.014	1.71 (1.10–2.67)	5.57	1	0.018

BBS: Berg balance score; FSS: fatigue severity scale; MMSE: mini mental state examination; FES: Fear of falling. Adjusted for sex, age, education, current EDSS, disease duration, and BMI.

**Table 5 jcm-09-03604-t005:** Adjusted logistic model performance.

	Pain	FSS	FES
Statistic	Value (95% CI)	Value (95% CI)	Value (95% CI)
AUC	0.97 (0.94–1.00)	0.96 (0.91–1.00)	0.99 (0.98–1.00)
Sensitivity	0.92 (0.75–0.99)	0.88 (0.70–0.98)	0.96 (0.82–1.00)
Specificity	0.77 (0.65–0.87)	0.93 (0.84–0.98)	1.00 (0.93–1.00)
Positive Predictive Value	0.63 (0.52–0.733)	0.85 (0.67–0.94)	1.00 (0.92–1.00)
Negative Predictive Value	0.96 (0.86–0.99)	0.95 (0.87–0.98)	0.98 (0.88–1.00)
Accuracy	0.82 (0.72–0.89)	0.92 (0.84–0.97)	0.99 (0.93–1.00)

FSS = fatigue severity scale; FES = fear of falling.

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
