# Peer review of "Higher Disease and Pain Severity and Fatigue and Lower Balance Skills Are Associated with Higher Prevalence of Falling among Individuals with the Inflammatory Disease of Neuromyelitis Optica Spectrum Disorder (NMOSD)"

_jcm, 2020, doi:10.3390/jcm9113604_

Round 1

Reviewer 1 Report

The authors of this work demonstrate significant more fails among pactients with NMOSD. Despite is not a relevant breakthrough, the study is well designed and conducted, methods are appropiate, the cohort is quite big (among 100 individuals/group) and the output might help in designing policies to prevent associated consequences of falls. For all these reasons I recommend this study to be published in teh JCM.

Please, consider reviewing language style (Discussion section: 'one might ask'...) and formating text and figures (it lloks at different sizes and styles to me).

Author Response

Dear Reviewer,

Thank you for the valuable comments. Please find attached the detailed point-by-point-response. 

Reviewer 2 Report

The authors have conducted a cross-sectional study on the risk of falling in patients with NMOSD, aiming to assess both the relative risk as compared to healthy controls and the predictors of falls in patients with NMOSD. The study is globally well-written and bears some novelty in the field. Morover, strenghts and limits are discussed thoroughly and honestly, which I really appreciate.

I have some observations:

Abstract: the authors state that EDSS had no predictive value and then, 2 lines below, that disease impairments including EDD predicted falling. This is obviously a mistake, please correct.

Introduction: on line 74, the authors state thatthey treated as “exploratory” the question of the extent to which typical NMOSD-related factors such as higher scores on the EDSS and for fatigue, and pain, together with lower balance skills might predict frequency of falls. I am not sure to understand what “exploraorty” means in this context. In what sense is this research question different from the previous two?

Samples, paragraph “persons with NMOSD”: on line 100, the age range is set between 18 and 50 years. Why were older people excluded. This may be a reasonable choice, but it should be justified. Line 107: why have the patients with a recent relapse been excluded? Line 108: why have the patients with anti-MOG disease been excluded? I remark that these choices need justification since a possible bias can be introduced.

Imaging analysis: I feel that the categorisation of brain imaging into normal or abnormal is too coarse, since we expect unspecific findings in many patients.

Statistical analyses: herein fall my main concerns. Since a confounder is defined as a variable which is associated both with the independent variables and the outcome (or dependent variable), I feel that the list of confounders is too long and not always justified. For example, I don’ts see how brain MRI might be a confounder. Moreover, if you consider EDSS as a confounder (again, why?) you cannot test it in a logistic regression.

Discussion: from line 82 to line 97, the author speculate on the benefits of physical acitivity on reducing the risk of falls. I wonder why they did not consider the habit to perform physical activity as a potential predictor in the present study?

Author Response

Dear Reviewer,

Thank you very much for all your kind efforts. Please find attached the detailed point-by-point-response.
